# Natural Images are More Informative for Interpreting CNN Activations than State-of-the-Art Synthetic Feature Visualizations

**Judy Borowski**[*], **Roland S. Zimmermann**[*], **Judith Schepers, Robert Geirhos,**
**Thomas S. A. Wallis**[†‡]**, Matthias Bethge**[‡]**, Wieland Brendel**[‡]
University of Tübingen, Germany

## Abstract

Feature visualizations such as synthetic maximally activating images are a widely used explanation method to better understand the information processing of convolutional neural networks (CNNs). At the same time, there are concerns that these visualizations might not accurately represent CNNs' inner workings. Here, we measure how much extremely activating images help humans in predicting CNN activations. Using a well-controlled psychophysical paradigm, we compare the informativeness of synthetic images by Olah et al. [45] with a simple baseline visualization, namely natural images that also strongly activate a specific feature map. Given either synthetic or natural reference images, human participants choose which of two query images leads to strong positive activation. The experiment is designed to maximize participants' performance, and is the first to probe *intermediate* instead of final layer representations. We find that synthetic images indeed provide helpful information about feature map activations ($82 \pm 4\%$ accuracy; chance would be $50\%$). However, natural images—originally intended to be a baseline—outperform these synthetic images by a wide margin ($92 \pm 2\%$ accuracy). The superiority of natural images holds across the investigated network and various conditions. Therefore, we argue that visualization methods should improve over this simple baseline.

## 1 Introduction

As Deep Learning methods are being deployed across society, academia and industry, the need to understand their decisions becomes ever more pressing and the interest in explainable Artificial Intelligence (XAI) is growing. Under certain conditions, a "right to explanation" is even required by law in the European Union [17, 19]. We here focus on the popular post-hoc explanation method (or interpretability method) of *feature visualizations via activation maximization*, also known as *input maximization* or *maximally exciting images*. First introduced by Erhan et al. [15] and subsequently improved by many others [34, 39, 37, 40, 42], these synthetic, maximally activating images seek to visualize features that a specific network unit, feature map or a combination thereof is selective for. However, feature visualizations are surrounded by great controversy: How accurately do they represent a CNN's inner workings? In this work, we focus on the question of how useful they are for humans.

On the one hand, many researchers are convinced that feature visualizations are interpretable [20] and that "features can be rigorously studied and understood" [48]. Over the past few years, extensive

---

[*]Joint first and corresponding authors: `first.last@bethelab.org`

[†]Current affiliation Amazon.com; this contribution is prior work

[‡]Joint supervision

2nd Workshop on Shared Visual Representations in Human and Machine Intelligence (SVRHM), NeurIPS 2020.

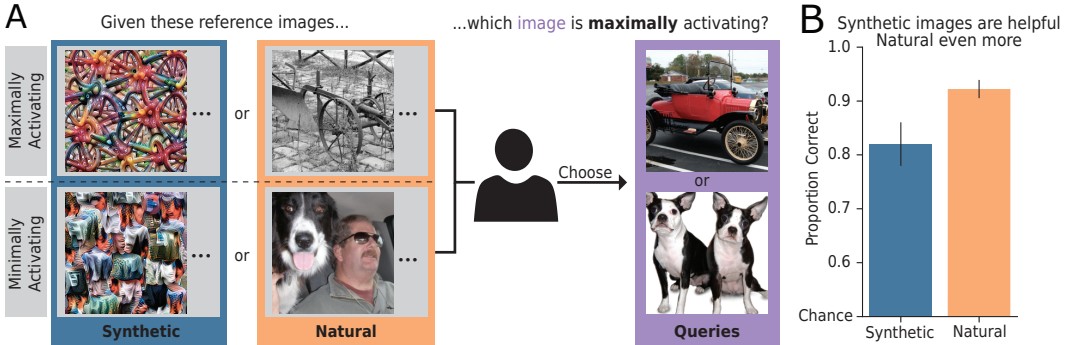

Figure 1: How useful are synthetic compared to natural images for interpreting neural network activations? **A: Human experiment.** Given extremely activating reference images (either synthetic or natural), a human participant chooses which out of two query images is also a strongly activating image. Synthetic images are generated via feature visualization [45]. **B: Core result.** Participants are well above chance for synthetic images—but even better when seeing *natural* reference images.

investigations to better understand CNNs are based on feature visualizations [48, 47, 8, 7], and the technique is being combined with other explanation methods [46, 9, 2, 23].

On the other hand, feature visualizations can be equal parts art and engineering as they are science: vanilla methods look noisy, thus human-defined regularization mechanisms are introduced. But do the resulting beautiful visualizations accurately show what a CNN is selective for? How representative are the well-interpretable, "hand-picked" [45] synthetic images in publications for the entirety of all units in a network, a concern raised by e.g. Kriegeskorte [29]? And what if the features that a CNN is truly sensitive to are imperceptible instead, as might be suggested by the existence of adversarial examples [55, 25]? Morcos et al. [36] even suggest that units of understandable features play a less important role in a network. Another criticism of synthetic maximally activating images is that they only visualize extreme features, while potentially leaving other features undetected that only elicit e.g. 70% of the maximal activation.

One way to advance this debate is to measure the utility of feature visualizations in terms of their helpfulness for *humans*. In this study, we therefore design well-controlled psychophysical experiments that aim to quantify the informativeness of the popular visualization method by Olah et al. [45]. Specifically, participants choose which of two natural images would elicit a higher activation in a CNN given a set of reference images that visualize the network selectivities. We use natural query images because real-world applications of XAI require understanding model decisions to natural inputs. To the best of our knowledge, our study is the first to probe how well humans can predict *intermediate* CNN activations. Our data shows that:

- Synthetic images provide humans with helpful information about feature map activations.
- Exemplary natural images are even more helpful.
- The superiority of natural images holds across the network and various conditions.

## 2 Related Work

Thanks to the growing number of explanation methods, significant progress has been made in recent years towards understanding CNNs for image data. Nevertheless, challenges remain and concern for example over-engineering. As such, the loss function and techniques to make the synthetic images look more interpretable are often discussed for feature visualizations [42]. Another critique is that interpretability research is not sufficiently tested against falsifiable hypotheses and rather relies too much on intuition [31].

In order to further advance XAI, scientists advocate different research directions. Besides the focus on developing additional methods, some researchers (e.g. Olah et al. [48]) promote the "natural science" approach, i.e. studying a neural network extensively and making empirical claims until falsification. Yet another direction is to quantitatively evaluate explanation methods. So far, only

decision-level explanation methods have been studied in this regard. Quantitative evaluations can either be realized with humans directly or with mathematically-grounded models as an approximation for human perception. Many of the latter approaches show great insights [24, 38, 16, 33, 58, 57]. However, a recent study demonstrates that metrics of the explanation quality computed without human judgment are inconclusive and do not correspond to the *human* rankings [6]. Additionally, Miller [35] emphasizes that XAI should build on existing research in philosophy, cognitive science and social psychology.

The body of literature on human evaluations of explanation methods is growing: Various combinations of data types (tabular, text, images), task set-ups and participant pools (experts vs. laypeople, on-site vs. crowd-sourcing) are being explored. However, these studies all aim to investigate final model decisions and do not probe intermediate activations like our experiments do. For a detailed table of related studies, see Appendix Sec. A.3. A commonly employed task paradigm is the "forward simulation / prediction" task, first introduced by Doshi-Velez and Kim [14]: Participants guess the model's computation based on an input and an explanation. As there is no absolute metric for the goodness of explanation methods, comparisons are performed within studies, often against baselines. According to the current literature, studies reporting positive effects of explanations [30] slightly outweigh those reporting inconclusive [4] or even negative effects [54].

To our knowledge, no study has yet evaluated the popular explanation method of feature visualizations and how it improves human understanding of intermediate network activations. This study therefore closes an important gap.

# 3   Methods

We perform two human psychophysical studies[4] with different foci (Experiment I ($N = 10$) and Experiment II ($N = 23$)). In both studies, the task is to choose the one image out of two natural query images (two-alternative forced choice paradigm) that the participant considers to be a strongly activating image given a set of reference images (see Fig. 2). Apart from image choices, we record participants' confidence levels and their reaction times. In order to gain insights on how intuitive participants find feature visualizations, their subjective judgments are collected in a separate task and a dynamic conversation (for details, see Appendix Sec. A.1.1 and Sec. A.2.6).

Figure 2: Example trial in psychophysical experiments. A participant sees minimally and maximally activating reference images for a certain feature map on the sides and has to select the image from the center that also strongly activates that feature map. The answer is given by clicking on the number according to the participant's confidence level. After each trial, the participant receives feedback which image was indeed the maximally activating one.

All design choices are made with two main goals: (1) allowing participants to achieve the *best performance possible* to approximate an upper bound on the helpfulness of the explanation method, and (2) gaining a *general* impression of the helpfulness of the examined method. To this end, we choose the natural query images from among those of lowest and highest activations ($\rightarrow$ best possible performance) and test many different feature maps across the network ($\rightarrow$ generality). For more details on the human experiment besides the ones below, see Appendix Sec. A.1.

In Experiment I, we focus on comparing the performance of synthetic images to two baseline conditions: natural reference images and no reference images. In Experiment II, we compare lay vs. expert participants as well as different presentation schemes of reference images. Expert participants qualify by being familiar or having practical experience with feature visualization techniques or at least CNNs. Regarding presentation schemes, we vary whether only maximally or both maximally and minimally activating images are shown; as well as how many example images of each of these are presented (1 or 9).

---

[4]Code is available at `https://github.com/bethgelab/testing_visualizations`.

Following the existing work on feature visualization [45, 46, 48, 47], we use an Inception V1 network [56] trained on ImageNet [12]. The synthetic images throughout this study are the optimization results of the feature visualization method by Olah et al. [45] with the spatial average of a whole feature map ("channel objective"). The natural stimuli are selected from the validation set of the ImageNet ILSVRC 2012 [52] dataset according to their activations for the feature maps of interest. Specifically, the images of the most extreme activations are sampled, while ensuring that each lay or expert participant sees different query and reference images. A more detailed description of the specific sampling process for natural stimuli and the generation process of synthetic stimuli is given in Sec. A.1.2.

## 4   Results

In this section, Fig. 3 (4) shows data from Experiment I (II). Additional figures as well as additional results (hand- vs. randomly picked, subjective impressions), can be found in the Appendix Sec. A.2. Error bars always denote two standard errors of the mean of the participant average metric.

### 4.1   Participants are Better, More Confident and Faster with Natural Images

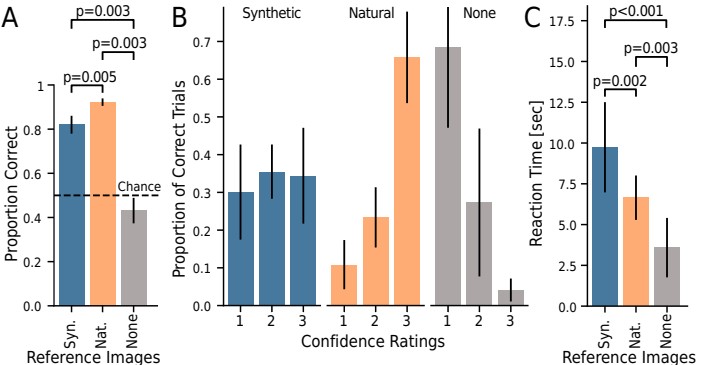

Synthetic images can be helpful: Given synthetic reference images generated via feature visualization [45], participants are able to predict whether a certain network feature map prefers one over the other query image with an accuracy of $82 \pm 4\%$, which is well above chance level ($50\%$) (see Fig. 3A). However, performance is even higher in what we intended to be the baseline condition: natural reference images ($92 \pm 2\%$). Additionally, for correctly answered trials, participants much more frequently report being highly certain on natural relative to synthetic trials (see Fig. 3B), and their average reaction time is approximately 3.7 seconds faster when seeing natural instead of synthetic reference images (see Fig. 3C). Taken together, these findings indicate that in our setup, participants are not just better overall, but also more confident and substantially faster when provided with natural images.

Figure 3: Participants are better, more confident and faster at judging which of two query images causes higher unit activation with natural than with synthetic reference images. **A: Performance.** Given synthetic reference images, participants are well above chance (proportion correct: $82 \pm 4\%$), but even better for natural reference images ($92 \pm 2\%$). Without references ("None"), participants are close to chance. **B: Confidence.** Participants are much more confident (higher rating = more confident) for natural than for synthetic images on correctly answered trials. **C: Reaction time.** For correctly answered trials, participants are on average faster when presented with natural than with synthetic reference images. The $p$-values correspond to Wilcoxon signed-rank tests.

### 4.2   For Expert and Lay Participants Alike: Natural Images are More Helpful

Explanation methods seek to explain aspects of algorithmic decision-making. Importantly, an explanation should not just be amenable to experts but to anyone affected by an algorithm's decision. We here test whether the explanation method of feature visualization is equally applicable to expert and lay participants (see Fig. 4A). Contrary to our prior expectation, we find no significant differences in expert vs. lay performance (RM ANOVA, $p = .44$, for details see Appendix Sec. A.2.2). Hence, extensive experience with CNNs is not necessary to perform well in this forward simulation task. In line with the previous main finding, both experts and lay participants are better in the natural than in the synthetic condition.

### 4.3 Additional Information Boosts Performance, Especially for Natural Images

Publications on feature visualizations vary in terms of how optimized images are presented [15, 9, 46, 59, 41, 45]. Naturally, the question arises as to what influence (if any) these presentation choices have. We here systematically compare presentations schemes along two dimensions (see Fig. 4B): the number of reference images (1 vs. 9) and the availability of minimally activating images (only Max vs. Min+Max). When just a single maximally activating image is presented (condition Max 1), natural images already outperform synthetic images ($73 \pm 4\%$ vs. $64 \pm 5\%$). With additional information along either dimension, performance improves both for natural as well as for synthetic images. The strongest boost in performance, however, is observed for natural reference images. In the Min+Max 9 condition, a replication of the result from Experiment I shown in Fig. 3A, natural images outperform synthetic images by an even larger margin ($91 \pm 3$ vs. $72 \pm 4\%$).

## 5 Discussion & Conclusion

Feature visualizations such as synthetic maximally activating images are a widely used explanation method, but it is unclear whether they indeed help humans to understand CNNs. Using psychophysical experiments with both expert and lay participants, we conduct the very first investigation of intermediate synthetic feature visualizations by Olah et al. [45]: Can participants predict which of two query images leads to a strong activation in a feature map, given extremely activating visualizations? Specifically, we shed light on the following questions:

(1.) *How informative are synthetic feature visualizations — and how do they compare to a natural image baseline?* In the prediction task, we find above-chance performance given synthetic feature visualizations, but to our own surprise, synthetic feature visualizations are systematically *less* informative than the simple baseline of natural strongly activating images. The subjective impressions of the interpretability of visualizations vary greatly between participants (see Appendix Sec. A.2.5). Interestingly, many synthetic feature visualizations contain regularization mechanisms to introduce more "natural structure" [45], sometimes even called a "natural image prior" [34, 44]. This raises the question: Are natural images all you need? One might posit that highly-activating natural (reference) images simply appear more similar to other highly-activating natural (query) images. While that might be true, feature visualizations are meant to explain feature map activations for natural images, and this is ultimately what real-world applications of XAI are concerned

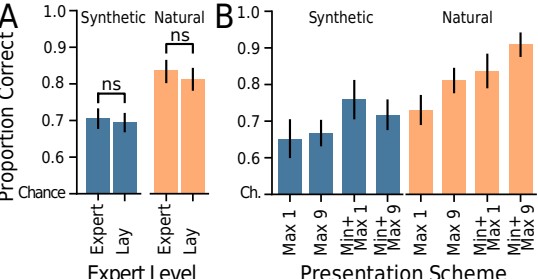

Figure 4: There is no evidence for large effects of expert level; however, performance does improve with additional information. **A: Expert level.** Both experts and lay participants perform equally well (RM ANOVA, $p = .44$), and consistently better on natural than on synthetic images. **B: Presentation scheme.** Presenting both maximally and minimally activating images simultaneously (Min+Max) and showing nine instead of one reference image tends to improve performance, especially for natural reference images. "ns" highlights non-significant differences.

with. On a different note, the independence of feature visualizations from the natural image manifold is often praised as an advantage because it supposedly reveals the unconstrained features used by a CNN. Again, while that may be true, the ever growing datasets and compute resources hold some promise that dataset examples may soon reveal similar - if not even better - insights.

(2.) *Do you need to be a CNN expert in order to understand feature visualizations?* To the best of our knowledge, our study is the first to compare the performances of expert and lay people when evaluating explanation methods. Previously, publications either focused on only expert groups [22, 30] or only laypeople [53, 4]. Our experiment shows no significant difference between expert and lay participants in our task—both perform similarly well, and even better on natural images: a replication of our main finding. Consequently, future studies may not have to rely on selected expert participants, but may leverage larger lay participant pools.

(3.) *What is the best way of presenting images?* Existing work suggested that more than one example [43] and particularly negative examples [28] enhance human understanding of data distributions.

Our systematic exploration of presentation schemes provides evidence that increasing the number of reference images as well as presenting both minimally *and* maximally activating reference images (as opposed to only maximally activating ones) improve human performance. This finding might be of interest to future studies aiming at peak performance.

**Caveats.** Despite our best intentions, a few caveats remain: The forward simulation paradigm is only one specific way to measure the informativeness of explanation methods, but does not allow us to make judgments about their helpfulness in other applications such as comparing different CNNs. Further, we emphasize that all experimental design choices were made with the goal to measure the best possible performance. As a consequence, our finding that synthetic reference images help humans predict a network's strongly activating image may not necessarily be representative of a less optimal experimental set-up with e.g. query images corresponding to less extreme feature map activations. Finally, while we explored one particular method in depth [45]; it remains an open question whether the results can be replicated for other feature visualizations methods.

**Future directions.** Besides using query images with more similar activations, future participants could be provided with more information, e.g. where a feature map is located in the network. Furthermore, it has been suggested that the combination of synthetic and natural reference images might provide synergistic information to participants [45], which could again be studied in our experimental paradigm. Additionally, further studies could explore single unit visualizations, combinations of units and different networks.

Taken together, our results highlight the need for thorough quantitative evaluations of feature visualizations and suggest that natural images provide a surprisingly challenging baseline for understanding CNN activations.

### Author Contributions

The initiative of investigating human predictability of CNN activations came from WB. JB, WB, MB and TSAW jointly combined it with the idea of investigating human interpretability of feature visualizations. JB led the project. JB, RSZ and JS jointly designed and implemented the experiments (with advice and feedback from RG, TSAW, MB and WB). The data analysis was performed by JB and RSZ (with advice and feedback from RG, TSAW, MB and WB). JB designed, and JB and JS implemented the pilot study. JB conducted the experiments (with help from JS). RSZ performed the statistical significance tests (with advice from TSAW and feedback from JB and RG). MB helped shape the bigger picture and initiated intuitiveness trials. WB provided day-to-day supervision. JB, RSZ and RG wrote the initial version of the manuscript. All authors contributed to the final version of the manuscript.

### Acknowledgments

We thank Felix A. Wichmann and Isabel Valera for helpful discussions. We further thank Alexander Böttcher and Stefan Sietzen for support as well as helpful discussions on technical details. Additionally, we thank Chris Olah for clarifications via `slack.distill.pub`. Also, we thank Matthias Kümmerer, Matthias Tangemann, Evgenia Rusak and Ori Press for helping in piloting our experiments, as well as feedback from Evgenia Rusak, Claudio Michaelis, Dylan Paiton and Matthias Kümmerer. And finally, we thank all our participants for taking part in our experiments.

We thank the International Max Planck Research School for Intelligent Systems (IMPRS-IS) for supporting JB, RZ and RG. We acknowledge support from the German Federal Ministry of Education and Research (BMBF) through the Competence Center for Machine Learning (TUE.AI, FKZ 01IS18039A) and the Bernstein Computational Neuroscience Program Tübingen (FKZ: 01GQ1002), the Cluster of Excellence Machine Learning: New Perspectives for Sciences (EXC2064/1), and the German Research Foundation (DFG; SFB 1233, Robust Vision: Inference Principles and Neural Mechanisms, TP3, project number 276693517).

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

# A  Appendix

## A.1  Details on methods

### A.1.1  Human Experiments

In our two human psychophysical studies, we ask humans to predict a feature map's strongly activating image ("forward simulation task", Doshi-Velez and Kim 14). Answers to the two-alternative forced choice paradigm (see Supplementary Material for screenshots) are recorded together with the participants' confidence level (1: not confident, 2: somewhat confident, 3: very confident). Time per trial is unlimited and we record reaction time. After each trial, feedback is given. A progress bar at the bottom of the screen indicates how many trials of a block are already completed. As reference images, either synthetic, natural or no reference images are shown. The synthetic images are the feature visualizations from the method of Olah et al. [45]. Trials of different reference images are arranged in blocks. Synthetic and natural reference images are alternated, and, in the case of Experiment I, framed by trials without reference images (see Fig. 5A, B). The order of the reference image types is counter-balanced across subjects.

The main trials in the experiments are complemented by *practice*, *catch* and *intuitiveness trials*. To avoid learning effects, we use different feature maps for each trial type per participant. Specifically, *practice trials* give participants the opportunity to familiarize themselves with the task. In order to monitor the attention of participants, *catch trials* appear randomly throughout blocks of main trials. Here, the query images are a copy of one of the reference images, i.e. there is an obvious correct answer (see Supplementary Material). This control mechanism allows us to decide whether trial blocks should be excluded from the analysis due to e.g. fatigue. To obtain the participant's subjective impression of the helpfulness of maximally activating images, the experiments are preceded (and also succeeded in the case of Experiment II) by three *intuitiveness trials* (see Supplementary Material). Here, participants judge in a slightly different task design how intuitive they consider the synthetic stimuli for the natural stimuli. For more details on the intuitiveness trials, see below.

At the end of the experiment, all expert participants in Experiment I and all lay (but not expert) participants in Experiment II are asked about their strategy and whether it changed over time. The information gained through the first group allows to understand the variety of cues used and paves the way to identify interesting directions for follow-up experiments. The information gained through the second group allowed comparisons to experts' impressions reported in Experiment I.

**Experiment I**  The first experiment focuses on comparing performance of synthetic images to two baselines: natural reference images and no reference images (see Fig. 5A). In total, 45 feature maps are tested: 36 of these are uniformly sampled from the feature maps of each of the four branches for each of the nine Inception modules. The other nine feature maps are uniformly hand-picked for interpretability from the Inception modules' pooling branch based on the appendix overview selection provided by Olah et al. [45] or based on our own choices. In the spirit of a *general* statement about

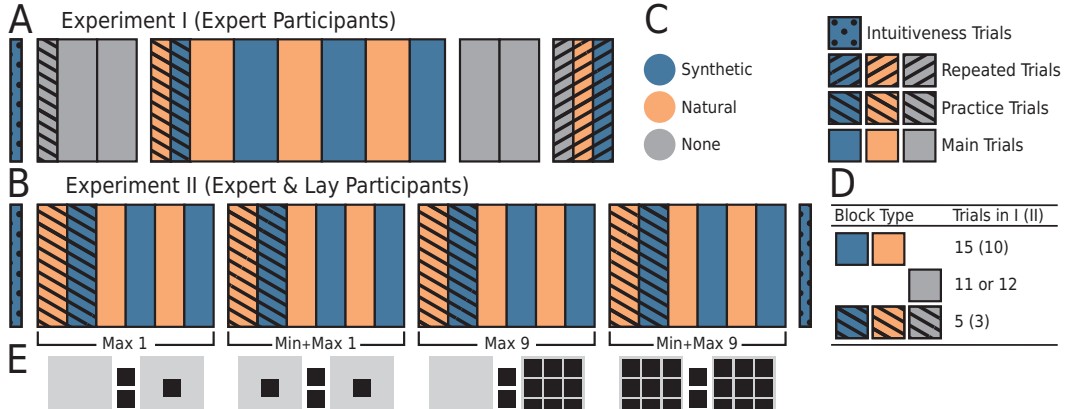

Figure 5: Detailed structure of the two experiments with different foci. **A: Experiment I.** Here, the focus is on comparing performance of synthetic and natural reference images to the most simple baseline: no reference images ("None"). To counter-balance conditions, the order of natural and synthetic blocks is alternated across participants. For each of the three reference image types (synthetic, natural and none), 45 relevant trials are used plus additional catch, practice and repeated trials. **B: Experiment II.** Here, the focus is on testing expert and lay participants as well as comparing different presentation schemes (Max 1, Min+Max 1, Max 9 and Min+Max 9, see **E** for illustrations). Both the order of natural and synthetic blocks as well as the four presentation conditions are counter-balanced across subjects. To maintain a reasonable experiment length for each participant, only 20 relevant trials are used per reference image type and presentation scheme, plus additional catch and practice trials. **C:** Legend. **D:** Number of trials per block type (i.e. reference image type and main vs. practice trial) and experiment. Catch trials are not shown in the figure; there was a total of 3 (2) catch trials per synthetic or natural main block in Experiment I (II). **E:** Illustration of presentation schemes. In Experiment II, all four schemes are tested, in Experiment I only Min+Max 9 is tested.

the explanation method, different participants see different natural reference and query images, and each participant sees different natural query images for the same feature maps in different reference conditions. To check the consistency of participants' responses, we repeat six randomly chosen main trials for each of the three tested reference image types at the end of the experiment.

**Experiment II** The second experiment (see Fig. 5B) is about testing expert vs. lay participants as well as comparing different presentation schemes[5] (Max 1, Min+Max 1, Max 9 and Min+Max 9, see Fig. 5E). In total, 80 feature maps are tested: They are uniformly sampled from every second layer with an Inception module of the network (hence a total of 5 instead of 9 layers), and from all four branches of the Inception modules. Given the focus on four different presentation schemes in this experiment, we repeat the sampling method four times without overlap. In terms of reference image types, only synthetic and natural images are tested. Like in Experiment I, different participants see different natural reference and query images. However, expert and lay participants see the same images. For details on the counter-balancing of all conditions, please refer to the Supplementary Material.

**Intuitiveness Trials** In order to obtain the participants' subjective impression of the helpfulness of maximally activating images, we add trials at the beginning of the experiments (and also at the end of Experiment II). The task set-up is slightly different (see Supplementary Material): Only maximally activating (i.e. no minimally activating) images are shown. We ask participants to rate how intuitive they find the explanation of the entirety of the synthetic images for the entirety of the natural images. Again, all images presented in one trial are specific to one feature map. By moving a slider to the right (left), participants judge the explanation method as intuitive (not intuitive). The ratings are recorded on a continuous scale from $-100$ (not intuitive) to $+100$ (intuitive). All participants see the same three trials in a randomized order. The trials are again taken from the hand-picked (i.e. interpretable)

---

[5]In pilot experiments, we learned that participants preferred 9 over 4 reference images, hence this "default" choice in Experiment I.

feature maps of the appendix overview in Olah et al. [45]. In theory, this again allows for the highest intuitiveness ratings possible. The specific feature maps are from a low, intermediate and high layer: feature map 43 of mixed3a, feature map 504 of mixed4b and feature map 17 of mixed 5b.

**Participants** Our two experiments are within-subject studies, meaning that every participant answers trials for all conditions. This design choice allows us to test fewer participants. In Experiment I, 10 expert participants take part (7 male, 3 female, age: 27.2 years, SD = 1.75). In Experiment II, 23 participants take part (of which 10 are experts; 14 male, 9 female, age: 28.1 years, SD = 6.76). Expert participants qualify by being familiar or having worked with convolutional neural networks and most of them even with feature visualization techniques. All subjects are naive with respect to the aim of the study. Expert (lay) participants are paid 15€ (10 €) per hour for participation. Before the experiment, all subjects give written informed consent for participating. All subjects have normal or corrected to normal vision. All procedures conform to Standard 8 of the American Psychological 405 Association's "Ethical Principles of Psychologists and Code of Conduct" (2016). Before the experiment, the first author explains the task to each participant and ensures complete understanding. For lay participants, the explanation is simplified: Maximally (minimally) activating images are called "favorite images" ("non-favorite images") of a "computer program" and the question is explained as which of the two query images would also be a "favorite" image to the computer program.

**Apparatus** Stimuli are displayed on a VIEWPixx 3D LCD (VPIXX Technologies; spatial resolution $1920 \times 1080$ px, temporal resolution $120 \, \text{Hz}$). Outside the stimulus image, the monitor is set to mean gray. Participants view the display from $60 \, \text{cm}$ (maintained via a chinrest) in a darkened chamber. At this distance, pixels subtend approximately $0.024°$ degrees on average ($41$ ps per degree of visual angle). Stimulus presentation and data collection is controlled via a desktop computer (Intel Core i5-4460 CPU, AMD Radeon R9 380 GPU) running Ubuntu Linux (16.04 LTS), using PsychoPy [50, version 3.0] under Python 3.6.

### A.1.2 Stimuli Selection

**Model** Following the existing work on feature visualization [45, 46, 48, 47], we use an Inception V1 network[6] (also known as "GoogLeNet") [56] trained on ImageNet [12, 52]. Note that the Inception V1 network used in the mentioned previous work slightly deviates from the original network architecture: The $3 \times 3$ branch of Inception module mixed4a only holds $204$ instead of $208$ feature maps. To stay as close as possible to the aforementioned work, we also use their implementation and trained weights of the network[7]. We investigate feature visualizations for all branches (i.e. kernel sizes) of the Inception modules and sample from layers mixed3a to mixed5b before the ReLU non-linearity.

**Synthetic Images from Feature Visualization** The synthetic images throughout this study are the optimization results of the feature visualization method from Olah et al. [45]. We use the channel objective to find synthetic stimuli that maximally (minimally) activate the spatial mean of a given feature map of the network. We perform the optimization using lucid 0.3.8 and TensorFlow 1.15.0 [1] and use the hyperparameter as specified in Olah et al. [45]. For the experimental conditions with more than one minimally (maximally) activating reference image, we add a diversity regularization across the samples. In hindsight, we realized that we generated 10 synthetic images in Experiment I, even though we only needed and used 9 per feature map.

**Selection of Natural Images** The natural stimuli are selected from the validation set of the ImageNet ILSVRC 2012 [52] dataset. To choose the maximally (minimally) activating natural stimuli for a given feature map, we perform three steps (for a visual illustration, see Supplementary Material): First, we calculate the activation of said feature map for all pre-processed images (resizing to $256 \times 256$ pixels, cropping centrally to $224 \times 224$ pixels and normalizing) and take the spatial average to get a scalar representing the excitability of the given feature map caused by the crop. Second, we order the stimuli according to the collected activation values and select the $(N_{stimuli} + 1) \cdot N_{batches}$ maximally (respectively minimally) activating images. Here, $N_{stimuli}$ corresponds to the number

---

[6]This network is considered very interpretable [46], yet other work also finds deeper networks more interpretable [5]. More recent work, again, suggests that "analogous features [...] form across models [...]," i.e. that interpretable feature visualizations appear "universally" for different CNNs [48, 49].

[7]`github.com/tensorflow/lucid/tree/v0.3.8/lucid`

of reference images used (either 1 or 9, see Fig. 5, **E**) and $N_{batches} = 20$ determines the maximum number of participants we can test with our setup. Third, we distribute the selected stimuli into $N_{stimuli} + 1$ blocks. Within each block, we randomly shuffle the order of the images. Lastly, we create $N_{batches}$ batches of data by selecting one image from each of the blocks for every batch.[8]

The reasons for creating several batches of extremely activating natural images are two-fold: (1) We want to get a *general* impression of the interpretability method and would like to reduce the dependence on single images, and (2) in Experiment I, a participant has to see different query images in the three different reference conditions. A downside of this design choice is an increase in variability. The precise allocation is done as follows: In Experiment I, the natural query images of the none condition were always allocated the batch with $batch\_nr = subject\_id$, the query and reference images of the natural condition were allocated the batch with $batch\_nr = subject\_id + 1$, and the natural query images of the synthetic condition were allocated the batch with $batch\_nr = subject\_id + 2$. The allocation scheme in Experiment II can be found in the Supplementary Material.

**Selection of Feature Maps**  The selection of feature maps used in Experiment I is shown in the Supplementary Material.

### A.1.3   Data Analysis: Significance Tests

All significance tests were performed with JASP [26, version 0.13.1]. For the analysis of the distribution of confidence ratings (see Fig. 3B) we used contingency tables with $\chi^2$-tests. For testing pairwise effects in accuracy, confidence, reaction time as well as intuitiveness data, we report Wilcoxon signed-rank tests with uncorrected p-values (Bonferroni-corrected critical alpha values with family-wise alpha level of $0.05$ reported in all figures where relevant). These non-parametric tests are preferred for these data because they do not make distributional assumptions like Normally-distributed errors, as in e.g. paired $t$-tests. For testing marginal effects (main effects of one factor marginalizing over another) we report results from repeated measures ANOVA (RM ANOVA), which does assume Normality.

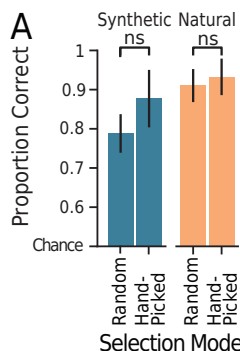

Figure 6: There is no significant performance difference between hand-picked feature maps selected for "interpretability" and randomly selected ones (Wilcoxon test, $p = 0.18$ for synthetic and $p = 0.59$ for natural reference images).

## A.2   Details on results

### A.2.1   Complementing Figures for Main Results

Fig. 7 complements the results and figures of Sec. 4. Here, all experimental conditions are shown.

### A.2.2   Details on Performance of Expert and Lay Participants

As reported in the main body of the paper, a mixed-effects ANOVA revealed no significant main effect of expert level ($F(1, 21) = 0.6$, $p = .44$, between-subjects effect). Further, there is no significant interaction with the reference image type ($F(1, 21) = 0.4, p = 0.53$), and both expert and lay participants show a significant main effect of the reference image type ($F(1, 21) = 230.2, p < 0.001$).

### A.2.3   Even for Hand-Picked Feature Visualizations, Performance is Higher on Natural Images

Often, explanation methods are presented using carefully selected network units, raising the question whether author-chosen units are representative for the interpetability method as a whole. Olah et al. [45] identify a number of particularly interpretable feature maps in Inception V1 in their appendix

---

[8]After having performed Experiment I and II, we realized a minor bug in our code: Instead of moving every 20th image into the same batch for one participant, we moved every 10th image into the same batch for one participant. This means that we only use a total of 110 different images, instead of 200. The minimal query image is still always selected from the 20 least activating images; the maximal query image is selected from the 89th to 109th maximally activating images - and we do not use the 109th to 200th maximally activating images.

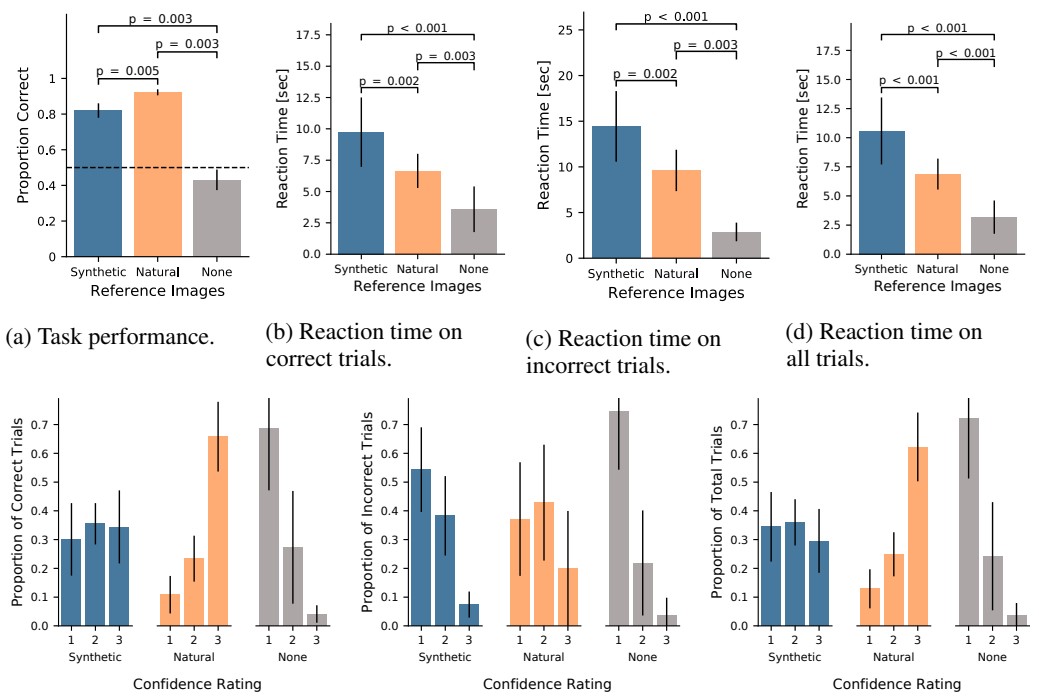

(a) Task performance.

(b) Reaction time on correct trials.

(c) Reaction time on incorrect trials.

(d) Reaction time on all trials.

(e) Confidence ratings on correctly answered trials.

(f) Confidence ratings on incorrectly answered trials.

(g) Confidence ratings on all trials.

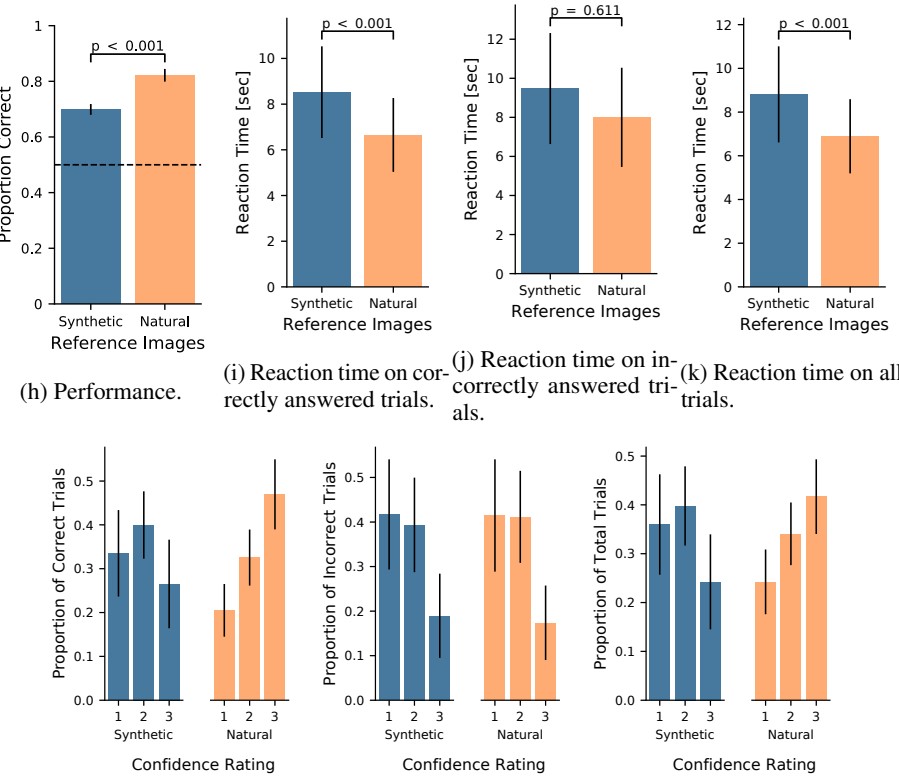

(h) Performance.

(i) Reaction time on correctly answered trials.

(j) Reaction time on incorrectly answered trials.

(k) Reaction time on all trials.

(l) Confidence ratings on correctly answered trials.

(m) Confidence ratings on incorrectly answered trials.

(n) Confidence ratings on all trials.

Figure 7: Results for Experiment I (II) are shown in the first (last) two rows: task performance (a, h), reaction times (b-d, i-k) and distribution of the confidence ratings (e-g, l-n). The $p$-values are calculated with Wilcoxon sign-rank tests.

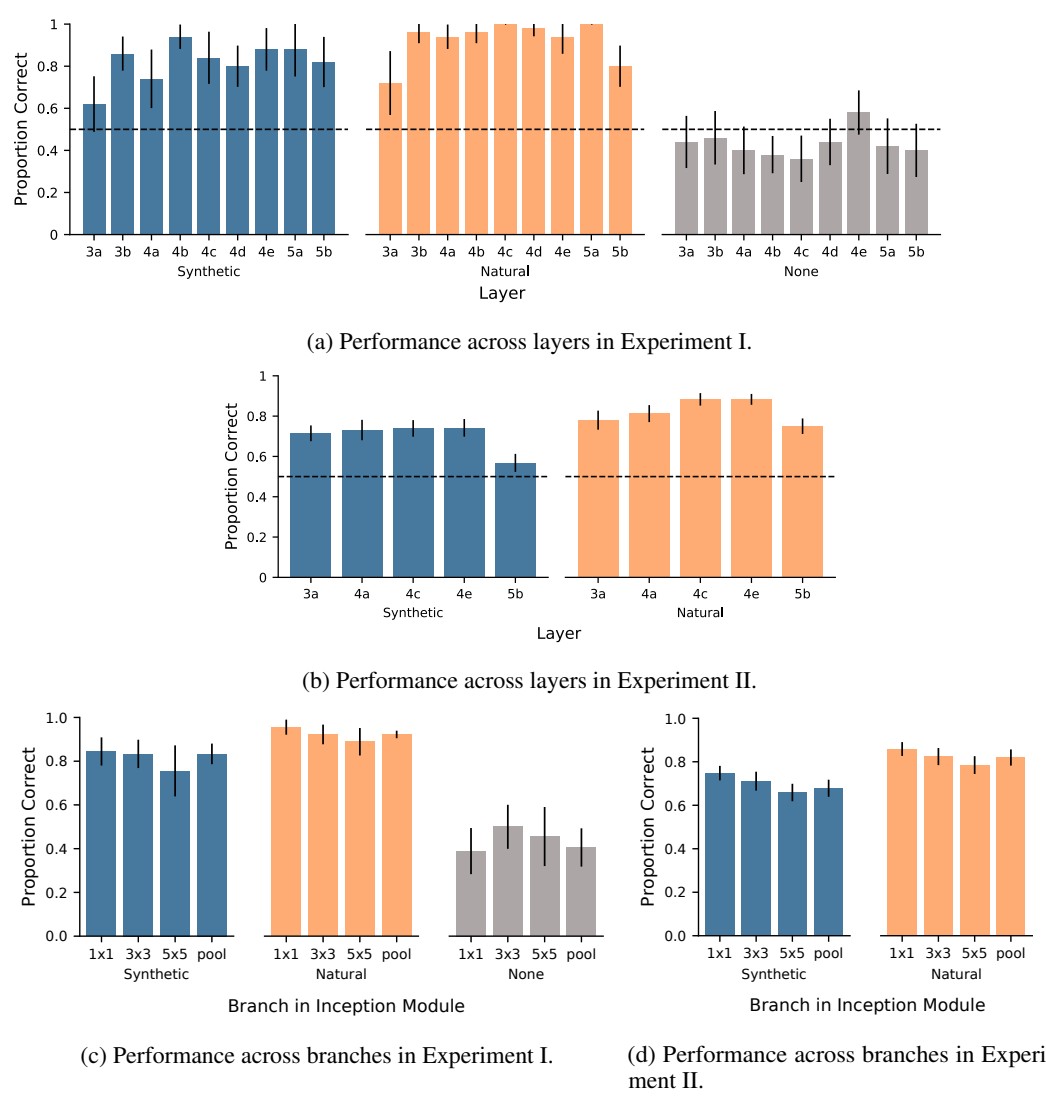

(a) Performance across layers in Experiment I.

(b) Performance across layers in Experiment II.

(c) Performance across branches in Experiment I.

(d) Performance across branches in Experiment II.

Figure 8: High performance across (a, b) layers and (c, d) branches of the Inception modules.

overview. When presenting either these hand-picked visualizations[9] or randomly selected ones, performance for hand-picked feature maps improves slightly (Fig. 6); however this performance difference is small and not significant for both natural (Wilcoxon test, $Z(9) = 27.5$, $p = 0.59$) and synthetic (Wilcoxon test, $Y(9) = 41$, $p = 0.18$) reference images. However, marginalizing over reference image type using a repeated measures ANOVA reveals a significant main effect of the feature map selection mode: $F(1,9) = 6.14$, $p = 0.035$. Therefore, while there may be a small effect of hand-picking feature maps, our data indicates that this effect, if present, is small. Nonetheless, consistent with the findings reported in the main paper, performance is higher for natural than for synthetic reference images *even on carefully selected hand-picked feature maps*.

### A.2.4 Natural Images are More Helpful Across a Broad Range of Layers

In our experiments, we also take a more fine-grained look at performance across different layers and branches of the Inception modules (see Fig. 8). Generally, feature map visualizations from

---

[9]All our hand-picked feature maps are taken from the pooling branch of the Inception module. As the appendix overview in Olah et al. [45] does not contain one feature map for each of these, *we* select interpretable feature maps for the missing layer mixed5a and mixed5b ourselves.

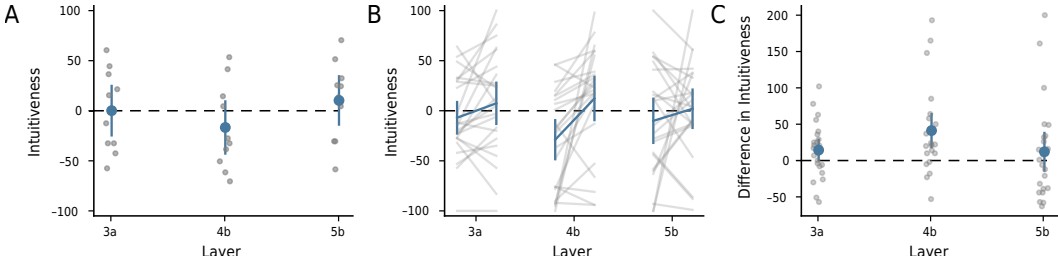

Figure 9: The subjective intuitiveness of feature visualizations varies greatly (see **A** for the ratings from the beginning of Experiment I and **B** for the ratings at the beginning and end of Experiment II). The means over all subjects yield a neutral result, i.e. the visualizations are neither un- nor intuitive, and the improvement of subjective intuitiveness before and after the experiment is only significant for one feature map (mixed4b). **C:** On average, participants found feature visualizations slightly more intuitive after doing the experiment as the differences larger than zero show. In all three subfigures, gray dots and lines show data per participant.

lower layers show low-level features such as striped patterns, color or texture, whereas feature map visualizations from higher layers tend to show more high-level concepts like (parts of) objects [32, 21, 18]. We find performance to be reasonably high across most layers and branches: participants were able to match both low-level and high-level patterns (despite not being explicitly instructed what layer a feature map belonged to). Again, natural images are mostly more helpful than synthetic images.

### A.2.5 Subjectively, Interpretability of Feature Visualizations Varies Greatly

While our data suggests that feature visualizations are indeed helpful for humans to predict CNN activations, we want to emphasize that our design choices aim at an upper bound on their informativeness. Another important aspect of evaluating an explanation method is the subjective impression. Besides recording confidence ratings and reaction times, we collect judgments on *intuitiveness trials* (see Supplementary Material), i.e. ratings of how intuitive feature visualizations appear for natural images. As Fig. 9A+B show, participants perceive the intuitiveness of synthetic feature visualizations for strongly activating natural dataset images very differently. Further, the comparison of intuitiveness judgments before and after the main experiments reveals only a small significant average improvement for one out of three feature maps (see Fig. 9B+C, Wilcoxon test, $p < .001$).

### A.2.6 Qualitative Findings

In a qualitative interview conducted after completion of the experiment, participants reported to use a large variety of strategies. Colors, edges, repeated patterns, orientations, small local structures and (small) objects were commonly mentioned. Most but not all participants reported to have adapted their decision strategy throughout the experiment. Especially lay participants from Experiment II emphasized that the trial-by-trial feedback was helpful and that it helped to learn new strategies. As already alluded to in the Discussion of the main paper, participants reported that the task difficulty varied greatly; while some trials were simple, others were challenging. A few participants highlighted that the comparison between minimally and maximally activating images was a crucial clue and allowed employing the exclusion criterion: If the minimally activating query image was easily identifiable, the choice of the maximally activating query image was trivial. This aspect motivated us to conduct an additional experiment where the presentation scheme was varied (Experiment II).

Furthermore, the interactive conversations painted a similar picture as the intuitiveness trials (see Sec. A.2.5): Some synthetic feature visualizations are perceived as intuitive while others do not correspond to understandable concepts. Nonetheless, four participants report that their first "gut feeling" for interpreting these reference images (as one participant phrased it) is more reliable. A few participants point out that the synthetic visualizations are exhausting to understand. Three participants additionally emphasize that the minimally activating reference images played an important role in their decision-making.

### A.2.7 High Quality Data as Shown by High Performance on Catch Trials

We integrate a mechanism to probe the quality of our data: In *catch trials*, the correct answer is trivial and hence incorrect answers might suggest the exclusion of specific trial blocks (for details, see Sec. A.1.1). Fortunately, very few trials are missed: In Experiment I, only two (out of ten) participants miss one trial each (i.e. a total of 2 out of 180 catch trials were missed); in Experiment II, five participants miss one trial and four participants miss two trials (i.e. a total of 13 out of 736 catch trials were missed) As this indicates that our data is of high quality, we do not perform the analysis with excluded trials as we expect to find the same results.

---

[8]Baseline condition.

[9]Metrics of explanation quality computed without human judgment are inconclusive and do not correspond to human rankings.

[10]Task has an additional "I don't know"-option for confidence rating.

[11]Comparison is only performed between methods but no absolute measure of interpretability for a method is obtained.

## A.3 Details on Related Work

| Paper | Analyzes Intermediate Features? | Explanation Methods Analyzed | Explanation helpful? | Results Confidence/Trust |
|---|---|---|---|---|
| **Ours** | yes | • Feature Visualization
• natural images[8]
• no explanation[8] | yes | • high variance in confidence ratings
• natural images are more helpful |
| Biessmann & Refiano (2019) | no | • LRP
• Guided Backprop
• simple gradient[8] | yes | • highest confidence for guided backprop[9] |
| Chu et al. (2020) | no | • prediction + gradients
• prediction[8]
• no information[8] | no | • faulty explanations do not decrease trust |
| Shen & Huan (2020) | no | • Extremal Perturb
• GradCAM
• SmoothGrad
• no explanation[8] | no | • - |
| Jeyakumar et al. (2020) | no | • LIME
• Anchor
• SHAP
• Saliency Maps
• Grad-CAM++
• Ex-Matchina | unclear[11] | • - |
| Alqaraawi et al. (2020) | no | • LRP
• classification scores
• no explanation[8] | yes | • confidence similar across conditions |
| Chandra-sekaran et al. (2017) | no | • prediction confidence
• attention maps
• Grad-CAM
• no explanation[8] | no | • - |
| Schmidt & Biessmann (2019) | no | • LIME
• custom method
• random/no explanation[8] | yes | • humans trust own judgement regardless explanations, except in one condition |
| Hase & Bansal (2020) | no | • LIME
• Prototype
• Anchor
• Decision Boundary
• combination of all 4 | partly | • high variance in helpfulness
• helpfulness cannot predict user performance |
| Kumaraku-lasinghe et al. (2020) | no | • LIME | yes | • fairly high trust and reliance |
| Ribeiro et al. (2018) | no | • LIME
• Anchor
• no explanation[8] | yes | • high confidence for Anchor
• low for LIME & no explanation |
| Alufaisan et al. (2020) | no | • prediction + Anchor
• prediction[8]
• no information[8] | partly | • explanations do not increase confidence |
| Dieber & Kirrane (2020) | no | • LIME | partly | • authors report that high hopes were lowered during challenging setup process, but good experience once everything was running |

| Paper | Experimental Setup | | | |
|---|---|---|---|---|
| | **Dataset** | **Task** | **Participants** | **Collected Data** |
| **Ours** | • natural images (ImageNet) | • CNN activation classification | • experts
• laypeople | • decision • confidence
• reaction time
• post-hoc evaluation |
| Biessmann & Refiano (2019) | • face images (Cohn-Kanade) | • 2-way classification[10] | • laypeople | • decision • confidence
• reaction time |
| Chu et al. (2020) | • face images (APPA-REAL) | • age regression | • laypeople | • decision • trust
• reaction time
• post-hoc evaluation |
| Shen & Huan (2020) | • natural images (ImageNet) | • model error identification | • laypeople | • decision |
| Jeyakumar et al. (2020) | • natural images (CIFAR-10)
• text (Sentiment140)
• audio (Speech Commands)
• sensory data (MIT-BIH Arrhythmia) | • preference for one out of two explanation methods | • laypeople | • decision |
| Alqaraawi et al. (2020) | • natural images (Pascal VOC) | • classification | • technical background (neither lay nor expert) | • decision
• confidence
• free answer on features |
| Chandra-sekaran et al. (2017) | • VQA (visualqa.org) | • model error identification
• regression | • laypeople | • decision |
| Schmidt & Biessmann (2019) | • book categories
• Movie reviews (IMDb) | • 9-/2-way classification | • laypeople | • decision
• reaction time
• trust |
| Hase & Bansal (2020) | • movie reviews (Movie Review)
• tabular (Adult) | • 2-way classification | • experts | • decision
• helpfulness rating
• explanation helpfulness |
| Kumaraku-lasinghe et al. (2020) | • tabular (Patient data) | • 2-way classification | • experts | • decision
• feature ranking
• satisfaction
• questionnaire |
| Ribeiro et al. (2018) | • tabular (Adult, rcdv) | • 2-way classification[10]
• VQA | • experts | • decision
• reaction time
• confidence |
| Alufaisan et al. (2020) | • tabular (COMPAS, Census Income) | • 2-way classification | • laypeople | • decision
• confidence
• reaction time |
| Dieber & Kirrane (2020) | • tabular (Rain in Australia) | • interview | • laypeople
• experts | • answers to how interpretable LIME output is |

Table 1: Overview of publications that evaluate explanation methods in human experiments. Note that the table already starts on the previous page and that the footnotes are displayed on page 17.

