# OpenReview forum: "Natural Images are More Informative for Interpreting CNN Activations than State-of-the-Art Synthetic Feature Visualizations"
_NeurIPS.cc/2020/Workshop/SVRHM — SVRHM@NeurIPS Poster_

### Official Review · AnonReviewer3 · 2020-10-28
**A human psychophysics based evaluation method for CNN explanation methods, clearly written with sound experiment design and result presentation**

**Rating:** 8
**Confidence:** 4

**Review:**

Summary of contributions:
In this study, the authors investigate how informative are the synthetic/natural maximally activating images in interpreting the activations of a deep neural network and perform multiple psychophysics experiments to identify which query image will show strong activation given some reference images. They find that both synthetic and natural images are highly informative, and surprisingly natural images seem to be much more informative than synthetic images.

Significance:
This study is relevant to interpreting the CNNs and provides an evaluation measure of different explanation methods such as synthetic maximally activating images. A similar evaluation approach could be applied to other CNN explanation methods.

Pros:
1. The paper is well written and the experiments are clearly explained.
2. The experiment design  is sound and focuses on the key questions about the information in maximally activating images i.e.  explanation quality evaluated using  human judgment
3. The results are reported with statistical tests supporting the evidence presented in the paper

Cons:
1.  The experiments are performed only using a single network InceptionV1

Suggestion to authors:
1. Although in supplementary, the authors show that natural images outperform synthetics images over different layers, it might be interesting to validate the result on another network.  The current study only investigates Inception V1 and therefore it raises the question of whether the results generalize to other types of networks.
2. This point is related to the previous one in the context that in order to investigate multiple networks, authors might consider using a few shot learning model as a proxy to human judgment and apply it to investigate how well the result generalizes across the networks. Then, based on these results the authors may consider selecting a few interesting models for psychophysics experiments to reduce the number of experiments for a large number of networks.

---

> ### Public Comment · ~Judy_Borowski1 · 2020-12-07
> **Answer to Reviewer 3**
>
> Dear Reviewer 3,
>
> Thank you for your positive and constructive feedback!
>
> We will answer your points by topic in chronological order:
>
> (1) “[...] it might be interesting to validate the result on another network.”
> We agree that this is an interesting future direction!
> We hypothesize that results would indeed look similar as the feature visualizations for other models also reveal similar features (see https://microscope.openai.com/models and “universality” claim in https://distill.pub/2020/circuits/zoom-in/).
>
> (2) Use a model as a proxy to human data
>
> That’s an interesting suggestion! We have indeed been discussing some ideas going in this direction, and we will consider extending our work along this line.

---

### Official Review · AnonReviewer1 · 2020-10-29
**Natural images are more informative for interpreting CNN activations than synthetic feature visualizations**

**Rating:** 8
**Confidence:** 4

**Review:**

## Summary

This paper presents experiments testing whether, given information about which visual information maximally activates an intermediate-layer CNN feature map, people are able to infer which of two natural images would strongly activate it. In one condition, the visual information about feature preference is in the form of synthetic feature visualizations, and in the other, it is natural images. Across experiments, the authors vary whether only maximally activating, or both minimally and maximally activating image(s) are shown, and how many of each (1-9) are presented. The behavioral task is a 2AFC in which participants indicate which image would be strongly activating, along with their level of confidence in this judgment. Additionally, the authors compare the judgments of experts (experienced with CNNs) and lay participants. The main findings are that people perform better on the task given natural reference images, though they are still able to perform significantly above chance given synthetic feature visualization reference images. Experts do not show an advantage over non-experts, and show the same pattern of performance (better given natural reference images). Finally, participants perform better given many reference images, and when shown both minimally- and maximally-activating reference images rather than just maximally-activating ones. Based on these experiments, the authors conclude that feature visualizations are informative for inferring which natural images strongly activate an intermediate-layer feature map, but that synthetic feature visualizations are less informative than strongly activating natural images, and this is true for experts and non-experts alike.

## Pros

The authors make a compelling case for quantifying the utility of explainability methods like synthetic feature visualization. The work is relevant to a broad range of communities.

The experiments are well designed to test the specific question of whether synthetic feature visualizations provide enough training data for people to classify strongly activating images above chance. The inclusion of the natural reference images is a useful baseline. The authors consider a number of variables, for example how many reference images, and of what type, maximize human task performance. Overall, the work has several interesting and novel take-aways.

The paper is well written and the methods are for the most part clearly described (minor caveats below) to a standard of replicability.

## Cons

The authors make the strong claim that "Natural images are more informative for interpreting CNN activations than synthetic feature visualizations" (the title). However, the paper largely ignores the main reason that feature visualization is thought to be useful: it provides information about *why* -- or what about -- an image drives activation. A visualization of a snout detector is more specific than an image of a dog. The current set of experiments does not address the difference in informativeness along this dimension, but there are a number of questions that could be asked. I hope that in follow-up work, the authors investigate experimental methods for probing this aspect of feature visualization.

I appreciate the general space constraints, but the authors should nevertheless include key details about the objective used for synthetic feature visualization in the main text, in particular the fact that they were optimizing a channel objective.

In Figure 1, it was somewhat difficult to parse what constituted a trial. By contrast, the format of the example trials provided in the Supplement was clear. The authors might consider using boxes to group the synthetic and natural image pairs, respectively.

The authors use the phrase "forward simulation" to describe the behavioral task. This descriptor strikes me as a misnomer: participants are trying to infer which input images would be more likely to elicit strong activation given information about feature preferences. This is not what I would take "forward simulation" to mean. I don't insist that the authors change this phrase, but wanted to provide feedback on my impression of it.

## Overall evaluation

On balance, this work serves as a valuable starting point for quantifying the utility of synthetic feature visualization as a tool for understanding CNN representations, and presents several results that will be of interest to the community. Nevertheless, the authors should be more circumspect in their conclusions since they fail to address a key raison d'etre for feature visualization as a tool, namely that feature visualization provides information about *why* --what about -- an image drives activation.

---

> ### Public Comment · ~Judy_Borowski1 · 2020-12-07
> **Answer to Reviewer 1**
>
> Dear Reviewer 1,
>
> Thank you for your detailed, positive and constructive feedback!
>
> (1) “The authors make the strong claim that "Natural images are more informative for interpreting CNN activations than synthetic feature visualizations" (the title).”
>
> We adjusted the title: “Natural Images are More Informative for Interpreting CNN Activations than State-of-the-Art Synthetic Feature Visualizations”
>
> (2) “[a feature visualization] provides information about why -- or what about -- an image drives activation. [...] I hope that in follow-up work, the authors investigate experimental methods for probing this aspect of feature visualization.”
>
> This is an excellent point! We are indeed already working on investigating this special aspect of feature visualizations. . For now, we added it explicitly in the manuscript in Section 5 “Discussion & Conclusion”, paragraph two.
>
> (3) “the authors should [...] include [...] the fact that they were optimizing a channel objective”
>
> We agree that an explicit definition of the natural image baseline is critical for understanding our study and, therefore, extended the last paragraph of Sec.3: We explain that we used “the spatial average of a whole feature map (“channel objective”)” for feature visualizations and provide details on how we selected natural images.
>
> (4) “The authors might consider using boxes to group the synthetic and natural image pairs, respectively.”
>
> In Fig. 1, panel A, the synthetic and natural reference images are highlighted by blue and orange boxes, respectively. These colors correspond to the bars in panel B.
>
> (5) The term “forward simulation" “to describe the behavioral task” might be misunderstood.
>
> We use this term because Doshi-Velez and Kim define it in their 2017-paper: "Forward simulation/prediction: humans are presented with an explanation and an input, and must correctly simulate the model’s output (regardless of the true output)."

---

### Official Review · AnonReviewer2 · 2020-10-29
**paper review**

**Rating:** 5
**Confidence:** 4

**Review:**

This paper covers the central question of, “How accurately do feature visualizations represent a CNN’s inner workings (or in short, how useful are they)?”. The authors probe at this question using a controlled psychophysical study to quantify how informative feature visualization technique from Olah et al are compared to natural images. Their study finds that synthetic images help humans predict the maximally activating image, but natural images are even more helpful.

My main concern is that the question of how accurately a feature visualization technique represents the inner workings of a CNN is actually different from the question of how useful it is for a human experimenter. There may be tension between how interpretable an explanation is vs. how correct it is. Just as an example, the work of Geirhos et al., 2019 (https://arxiv.org/abs/1811.12231) suggests that ImageNet-trained models are biased towards texture rather than shape in order to make classifications - now we would want the maximally activating image to reflect this! Say we have an ImageNet-trained neural network and we are interested in how it classifies Golden Retrievers, and in this hypothetical example the network actually detects fur features (texture) rather than full dog features (shape). In this example, the synthetic feature visualizations might look like fur texture, but using natural images from ImageNet as stimuli for the task outlined in this paper would result in only examples of full dogs as possible reference images (because there are no fur pictures or very zoomed in dog pictures in ImageNet). While these reference images may help a person guess the right query image (which will also be a full picture of a dog, because it is taken from ImageNet) faster and more accurately, it may also lead the experimenter to believe that the network performs its classifications by looking for full-dog features (when in reality the network is looking for fur texture). Understanding the network via natural images in this hypothetical scenario is therefore perhaps useful for humans, but not necessarily accurate to the CNN’s inner workings.

Further, I find the central result of the paper - that humans are both faster and more accurate on natural images - not very surprising. Just as neural networks have trouble performing tasks on out-of-distribution data, we as humans are trained on natural images and I can imagine it is harder for us to interpret these images synthesized via gradient ascent.

Pros:
- The work is very original, to the best of my knowledge
- I think the finding that experts perform just as well as laypeople on this task is indeed very interesting
- The sub-study on how to best present images (maximally activating and minimally activating images) is also interesting!
- The design of the task from a psychology standpoint looks sound


Cons
- I personally think the main result that people are better with natural images is extremely obvious, for reasons mentioned above. - However if it is non-obvious I would love to see more of that intuition covered in the text
- I don’t understand the implications of this work for the machine learning community. There are problems with using natural images only to try and understand the inner workings of a CNN, for example the shape vs. texture example described above. Similarly, given a neural network with no knowledge of what classes it was trained on, it may be impossible to find a useful maximally activating natural image for this network. Where would you even begin the search for images?

For the reasons outlined I feel borderline on this work. Hopefully the other reviewers can also provide more insight.

---

> ### Public Comment · ~Judy_Borowski1 · 2020-12-07
> **Answer to Reviewer 2**
>
> Dear Reviewer 2,
>
> Thank you for your detailed and constructive review.
>
> We will answer your points by topic in chronological order:
>
> (1) “My main concern is that the question of how accurately a feature visualization technique represents the inner workings of a CNN is actually different from the question of how useful it is for a human experimenter.”
>
> We reformulated the ending of the introduction, paragraph 1 such that the reader knows which questions we address: “However, feature visualizations are surrounded by great controversy: How accurately do they represent a CNN's inner workings? In this work, we focus on the question of how useful they are for humans.”
>
>
> (2) “In this [dog fur vs. shape] example, the synthetic feature visualizations might look like fur texture, but using natural images from ImageNet as stimuli for the task outlined in this paper would result in only examples of full dogs as possible reference images (because there are no fur pictures or very zoomed in dog pictures in ImageNet).”
>
> We agree that feature visualizations can be a useful tool to understand the inner workings in DNNs and that an advantage is their independence of correlations in natural images. To make this clearer, we adjusted our manuscript at two locations: In the Discussion & Conclusion section (Sec. 5), we now added in the first paragraph that “the independence of feature visualizations from the natural image manifold is often praised as an advantage because it supposedly reveals the unconstrained features used by a CNN.” We further hypothesize that “the ever growing datasets and compute resources [may] reveal similar - if not even better - insights.”
>
> (3) “I find the central result of the paper - that humans are both faster and more accurate on natural images - not very surprising. Just as neural networks have trouble performing tasks on out-of-distribution data, we as humans are trained on natural images and I can imagine it is harder for us to interpret these images synthesized via gradient ascent.”
>
> We understand the perspective that humans might perform better on natural reference images as they are used to this kind of images. However, it’s not entirely clear to us as to whether feature visualisations really are at a disadvantage in our experiment: The explanation method is not constrained by the natural image manifold and may thus work out more clearly to what the feature maps respond to, which in turn should help in our task. As we point out in the discussion section, the kind of images that explanation methods will be applied to in the real world will also be natural. Therefore, we consider our choice of testing humans on natural images as the most reasonable starting point.
>
> (4) “I don’t understand the implications of this work for the machine learning community”
>
> One goal of XAI research is to develop interpretability methods that help humans understand AI algorithms better. As Leavitt and Morcos (2020) point out ‒ and we now added to the related work section ‒ explanation methods (including feature visualizations) suffer from an “over-reliance” on intuition and should more extensively be tested against falsifiable hypotheses.
> Our work is a step in that direction: For the first time, the helpfulness of feature visualizations for humans is evaluated. Our main finding that natural images provide more helpful information than the synthetic images (of the one specific feature visualization method tested)  in our feedforward simulation task suggests that other interpretability methods should improve over such baselines.
>
> (5) “Similarly, given a neural network with no knowledge of what classes it was trained on, it may be impossible to find a useful maximally activating natural image for this network. Where would you even begin the search for images?”
>
> We agree: it might be problematic if a visualization method could only be applied to networks where we know exactly which classes it was trained on. Fortunately, this is not the case for the natural image baseline, where we don’t need to know anything about the training classes. The only requirement is  having access to data that comes from a similar distribution as the ones the network was originally trained on. We argue that in practice if one is interested in understanding a network, one is interested in understanding its behavior with respect to a collection of data. From this set of images, one can then simply select those that are more strongly activating than the others.

---

### Public Comment · ~Judy_Borowski1 · 2020-12-07
**Authors’ Update on Camera-Ready Version**

We would like to thank all three reviewers for their valuable and constructive reviews. We are happy they consider our work “original” (R2), “relevant to a broad range of communities” (R1), “well written” and “exceptionally clear” (R3).
We updated the manuscript based on your constructive feedback and suggestions and hope that these changes further improve the clarity.

---

### Decision · Program_Chairs · 2020-11-02

Accept (Poster)